# A Food-Based Science, Technology, Engineering, Arts, and Mathematics Learning Program May Improve Preschool Children’s Science Knowledge and Language Skills in Rural North Carolina

**DOI:** 10.3390/nu17091523

**Published:** 2025-04-30

**Authors:** Virginia C. Stage, Jocelyn B. Dixon, Pauline Grist, Qiang Wu, Archana V. Hegde, Tammy D. Lee, Ryan Lundquist, L. Suzanne Goodell

**Affiliations:** 1Department of Agricultural & Human Sciences, College of Agriculture and Life Sciences, North Carolina State University, Raleigh, NC 27695, USA; jocelyn_dixon@ncsu.edu (J.B.D.); pgrist@ncsu.edu (P.G.); rklundqu@ncsu.edu (R.L.); 2Department of Public Health, East Carolina University, Greenville, NC 27834, USA; wuq@ecu.edu; 3Department of Human Development & Family Science, College of Health and Human Performance, East Carolina University, Greenville, NC 27834, USA; hegdea@ecu.edu; 4Department of Mathematics, Science, & Instructional Technology Education, College of Education, East Carolina University, Greenville, NC 27834, USA; leeta@ecu.edu; 5Department of Food, Bioprocessing and Nutrition Sciences, College of Agriculture and Life Sciences, North Carolina State University, Raleigh, NC 27695, USA; lsgoodel@ncsu.edu

**Keywords:** STEAM, food-based learning, science, Head Start, intervention, Veggie Meter^®^, preschool

## Abstract

**Background/Objectives**: Early childhood represents a sensitive period for developing positive dietary preferences and important school readiness skills. However, few evidence-based programs leverage opportunities to support children’s development in both areas. Our study aimed to assess the preliminary effects of multi-level, teacher-led *More PEAS Please!* on Head Start children’s (3–5 years old) science knowledge, development of academic language, fruit-and-vegetable (FV) liking, and dietary quality. **Methods**: In this pilot study, we used a repeated-measure research design to assess child-level outcomes. Trained teachers implemented 16 food-based science-learning activities. We assessed child outcomes using validated measures of science knowledge, academic language, FV liking, and dietary quality (Veggie Meter^®^). We used linear mixed models to examine changes from the baseline to post intervention. Fixed effects included age, sex, and race/ethnicity, while the center was treated as a random effect. **Results**: A total of 273 children were enrolled in the study. The children were mostly male (51.6%), Black/African American (82.1%) and, on average, 3.94 (SD = 0.70) years old. The children demonstrated significant improvements in science knowledge (T1 M = −0.01, SD = 0.82; T4 M = 0.33, SD = 0.90; 95% CI [0.17, 0.50]; *p* < 0.001) and vocabulary (T1 M = 14.4, SD = 4.5; T4 M = 16.7, SD = 5.3; 95% CI [1.4, 3.3]; *p* < 0.001). The children’s dietary quality improved from the baseline, but the changes were not significant. **Conclusions**: The findings suggest that the intervention may support improvements in science knowledge and academic vocabulary among preschool-aged children. We theorize a longer intervention with additional FV exposures may be needed to observe significant dietary changes. Future research should evaluate program effects with a comparison group.

## 1. Introduction

Children’s early learning experiences lay the foundation for the rapid development of sensory, language, cognitive, and health behaviors unparalleled to those in any other stage of life [1,2,3]. Specifically, preschool (ages 3–5) is a critical time for neural development and the formation of behavioral patterns related to cognitive and health-related outcomes [4,5]. It is also when many children transition from home to an early care and education (ECE) center for the first time, often spending over 30 h per week in ECE settings [6,7]. Thus, for many children, a significant influence on their health and academic development occurs outside of the home in the ECE environment [4,8]. For almost one million children in the US, this ECE setting is Head Start, the federally funded preschool program serving children and families with limited resource backgrounds since 1965 [9]. Head Start strives to provide well-rounded care, including meeting the emotional, social, health, nutritional, and psychological needs of the children and families they serve [9]. 

Although the services and education that Head Start provides are essential to establish positive academic and health behaviors in preschool-age children, approximately one in three children entering Head Start are categorized as overweight or obese, with a BMI at or above the 85th percentile [10]. Additionally, 27% of preschool-age children do not consume the daily recommended number of vegetables [11], with children from low-resource families having an even lower intake of healthy foods, potentially because of limited access [12,13]. The high rates of childhood overweight and obesity, coupled with low rates of healthy food consumption, are alarming because the habits established during the preschool years (3–5 years old) are suggested to impact long-term health status into adulthood [1,14,15,16,17]. Not only that but also children with poorer health outcomes often experience lower academic performance, higher rates of disability, reduced job opportunities, and limited engagement in their communities as they transition into adulthood [18,19].

However, children’s positive experiences with their ECE teachers in health and academics can have significant positive long-term effects. Related to health, positive experiences in preschool may lower children’s risk for diet-related chronic illnesses, such as adult obesity, hypertension, type II diabetes, and cardiovascular disease [20,21,22]. Academically, early positive experiences can positively shape children’s long-term academic success, including increasing children’s interest and self-efficacy in rapidly expanding fields, such as Science, Technology, Engineering, Arts, and Mathematics (STEAM) [22,23,24,25,26,27,28]. Therefore, ECE teachers are in a critical position to positively influence children’s long-term outcomes in both health and academics [22,29,30,31]. One study quantified this potential impact, estimating a 7.3:1 annualized rate of return in long-term societal benefits [22].

One key method to impact children’s exposure to healthy foods as well as academic outcomes is food-based learning (FBL) [32,33,34,35]. Food-based learning has been previously defined as the “use of food as a tool to provide repeated exposure to improve children’s dietary behaviors and/or academic learning related to knowledge (e.g., science, mathematics, and literacy) and/or skills (e.g., gross motor, fine, and physical)” [32]. Instead of leaving the study of food for mealtimes, where children feel an inherent pressure to “eat” foods rather than explore them [36], FBL ‘turns the table’ and provides children with the opportunity to explore foods freely outside of mealtimes through STEAM exploration, including reading, gardening, scientific experimentation, and art [37,38,39]. Children’s exposure to healthy foods in a low-pressure environment, such as FBL, is essential, as 8–12 exposures to a novel food are necessary to positively change the preference or consumption of that food, with additional exposures needed for known, not-yet-preferred foods [40,41] and/or for children experiencing sensory-processing disorder or other medical diagnoses that impact eating [42,43]. Therefore, FBL can be an avenue to not only increase food exposures but also teach fundamental academic concepts simultaneously (e.g., life sciences and literacy) [32,33].

Previous interventions that introduced healthy foods to preschool classrooms through FBL have shown potential in increasing fruit and vegetable (FV) consumption among children [33,35]. One such program, *Together, We Inspire Smart Eating* (WISE), is an 8-month nutrition education curriculum designed to enhance children’s exposure to FVs through weekly hands-on activities using foods available in both school and home settings [35]. The curriculum integrates seamlessly into regular classroom routines, such as circle time, and connects with other subjects, including mathematics. For example, a WISE activity involves children creating skewers with spinach, tomatoes, and mozzarella cheese to form patterns, teaching a basic concept in mathematics. The effectiveness of WISE was demonstrated by significant improvements in vegetable intake, as observed in parental reports of food frequency questionnaires at the study’s conclusion (Mean = 3.44, SD = 1.23). Additionally, the study utilized resonance Raman spectroscopy (RRS), which measures blood carotenoid levels, a marker of FV intake. The results showed that children in the WISE program had notably higher RRS scores at the end of the intervention compared to those at the beginning (t(263) = −0.08, *p* < 0.039) [35]. A subsequent study by Bayles et al. further supports these findings, using seven hands-on STEAM FBL activities over four months [33]. This intervention included nine target vegetables, such as broccoli, cauliflower, and sweet potatoes, selected based on children’s prior exposure and their potential impact on the skin carotenoid status (SCS). Each session, lasting 15–20 min, combined circle time discussions with a hands-on activity that incorporated science, math, or language skills. The children had the chance to taste and explore the featured vegetable at the end of each lesson. An example activity from this study involved children steaming broccoli to explore the green pigment chlorophyll. The outcomes revealed that although both the intervention and comparison groups’ SCSs decreased between the baseline and post testing, the children in the intervention group had significantly higher SCSs at post testing compared to the comparison group, with a time-by-group interaction significant for the change in the SCSs (F 1,77 = 3.98; *p* = 0.02, r = 0.10) [33,35].

However, neither of these studies reported outcomes focused on children’s academic learning related to knowledge or skills (e.g., science, mathematics, and literacy/language). Although integrating FBL with academic learning is recognized as a strategy to address teaching challenges [32,33,34], there appears to be a significant research gap, specifically in investigating the effect of FBL on young children’s learning outcomes.

### Study Aims

This study aimed to assess the preliminary effects of *More PEAS Please!* on Head Start children’s (3–5 years old) science knowledge, academic language development, FV liking, and dietary quality. PEAS stands for Preschool Education in Applied Sciences. We hypothesized that children would demonstrate significant improvements in science knowledge, academic language development, FV liking, and dietary quality. Researchers used validated tools to measure the intervention’s preliminary effects on child-level outcomes.

## 2. Materials and Methods

### 2.1. More PEAS Please! Program Description

*More PEAS Please!* is a multi-component intervention designed to improve the quality of children’s early science-learning experiences in the preschool classroom by supporting changes in teachers’ instructional practices. This intervention aimed to improve children’s science knowledge, academic language development, FV liking, and dietary quality through early exposure to healthy foods and high-quality science-learning environments. We designed *More PEAS Please!* to be implemented over a full school year, with initial teacher training occurring in pre-service training. The program consisted of teachers’ professional development and implementation of food-based science-learning activities in the classroom. 

#### 2.1.1. Teacher Professional Development

Teacher professional development components included a one-day kick-starter workshop; six interactive, on-demand online learning modules; and center-based learning communities (LCs). The participating teachers attended a one-day kick-starter workshop held during pre-service training (in August, before the school year started). This workshop acquainted teachers with the program’s mission and objectives, delivered hands-on learning experiences based on evidence-based teaching practices, and established center-based LCs. At the end of the workshop, the participants received all the necessary materials, books, and detailed instructions needed for the effective implementation of the intervention.

Following the workshop, the teachers participated in ongoing professional development through six interactive, on-demand online learning modules. These modules highlighted the essential elements of effective teacher training, including active learning through practice, feedback, reflection, alignment with Head Start program goals, and a focus on child learning outcomes [44]. The modules were specifically designed to aid teachers in implementing classroom learning experiences. Finally, we facilitated the formation and ongoing support of center-based LCs, focusing on the adoption of new teaching strategies and creating a supportive environment to address educational challenges. We also conducted two virtual check-ins, one in fall and the other in spring, with each center-based team to ensure ongoing support and guidance.

#### 2.1.2. Classroom Intervention (Model Science-Learning Activities)

To enhance teachers’ training and ensure the practical application of new skills, our team of transdisciplinary faculty, staff, and Head Start partners developed 16 model science-learning activities using the well-established understanding-by-design (UBD) framework [45]. The model science-learning activities were organized into four thematic life science units: (1) living and non-living things, (2) seeds, (3) plants, and (4) plant parts (Table 1). Life science was chosen for these units because it offers a strong foundation for hands-on, inquiry-based learning, and many teachers and children have prior experiences in this area, making it an easier entry point to build teachers’ science knowledge, interest, and confidence while also supporting children’s science and language development. Additionally, as preschoolers grow, inquiry-based life-science-learning helps them to apply scientific thinking to make informed choices about their health and the environment [46,47,48].

Life science units integrated FBL with the development of children’s science and language skills and were suitable for both large group settings, like circle time, and smaller, center-based group activities. Following the *Preschool Cycle of Discovery* (Figure 1, Table 2), each of the four units consisted of four hands-on activities (16 in total). Across all the units, each activity emphasized sensory-focused learning by incorporating one or more target vegetables—carrots, tomatoes, spinach, and peas [38]. Teachers provided children with food exposures through hands-on observations, viewing color images of the foods in different forms, reading storybooks that featured the foods, and taste testing.

#### 2.1.3. More PEAS Please! Learning Standard Alignment

During the development process, we aligned model science-learning activities with the next-generation science standards (NGSS) [48] and Head Start’s Early Learning Outcomes Framework [9]. This alignment ensured that the program adhered to established educational standards while tackling common challenges identified by teachers [10,34,35,49,50]. Although the NGSS does not specifically target preschool education, classroom learning strategies and hands-on activities were designed to align with these standards to demonstrate how ECE teachers can foster school readiness in preschoolers. We achieved this by integrating disciplinary core ideas (Table 1) and science and engineering practices (Table 2) into developmentally appropriate learning experiences.

#### 2.1.4. More PEAS Please! Child-Level Theoretical Framework

Our driving theoretical model was derived from Social Cognitive Theory [51] and Johnson’s two-stage model of influences on children’s vegetable consumption [52] (Figure 2). The featured constructs, particularly repeated exposure [38,52], engaging senses [38], and role modeling [52,53], are consistently found to be effective in improving children’s dietary quality. We hypothesize that a school-based intervention providing teachers with training on how to implement evidence-based practices across health (nutrition), cognition (science), and language-learning domains will improve children’s dietary quality and school readiness in the areas of cognition and language.

### 2.2. Methods

#### 2.2.1. Research Design

We conducted this repeated-measure pilot study in collaboration with a partnering Head Start organization that operated five centers (twenty-three classrooms) across three rural North Carolina (NC) counties. We used validated and researcher-developed evaluation tools to measure preliminary child outcomes. East Carolina University’s Institutional Review Board (#21-001272) reviewed and approved all the study materials on 30 June 2021.

#### 2.2.2. Participant Recruitment

All the children aged 3–5 years, English and Spanish speaking, who were enrolled in classrooms with participating teachers, were eligible to participate. Teachers employed by the partnering Head Start organization were automatically eligible to participate in the program. We have reported teacher-level outcomes elsewhere [54]. Children with disabilities or developmental delays that interfered with their ability to respond to assessment tools were not included in data collection activities, but they benefited from the intervention’s classroom activities. To recruit child participants, we informed the families of children enrolled in participating Head Start centers about the study in two ways: (1) during the annual school registration in July 2021 and (2) via letters sent home at the start of the school year in August 2021. In the letter, we included (1) information detailing the study, (2) a waiver of parental permission form, and (3) optional surveys about their child, with a website link provided for online completion. Only the families who opted out of the study were required to sign and return the waiver of parental permission form.

#### 2.2.3. Data Collection

We collected child-level data throughout the 2021–2022 school year at four time points: the baseline (T1; September–October 2021), midpoint 1 (T2; December 2021), midpoint 2 (T3; February 2022), and post intervention (T4; April–May 2022). The measures of FV liking and dietary quality were the only ones assessed at the midpoints. We assessed children with evaluation tools individually at a low distraction, center-based location identified by each center director. We only collected data from the children who did not have a signed waiver on file and who readily gave assent on the day of the assessment. The children provided assent to participate by agreeing verbally and physically to participate in the data collection process.

##### Lens on Science (at the Baseline and Post Intervention)

The researchers used Lens on Science, a validated measure of science school readiness, to assess science knowledge [55] at the baseline and post intervention. A panel of experts in early childhood science independently evaluated each item to assess its developmental appropriateness and alignment with the NGSS. The items were pilot tested with children enrolled in Head Start programs to determine their effectiveness in distinguishing varying levels of understanding within this population. The correlation between each item and the ability estimate exceeded 0.20 for 87% of the items and surpassed 0.30 for 65% of the items. These results indicate that most items effectively differentiated between levels of ability and contributed to measuring a common underlying construct within the assessment [55]. For our study, the Lens on Science research team adapted the tool to an 80-item tool focused on life science concepts, reducing the administration time to approximately 15–20 min. A trained researcher monitored the administration of the tool to individual assenting/consenting preschoolers enrolled at one center. We administered the assessment using a touchscreen tablet or laptop, ensuring that the children received audio instructions and interactive practice before testing. The tool was available in both English and Spanish.

##### Academic Vocabulary (at the Baseline and Post Intervention)

To assess language development, a criterion-referenced, speech- and language-pathologist-developed vocabulary assessment was used at the baseline and post intervention [56]. The assessment included 40 novel words presented through images aligned with life science content and took approximately 25 min to administer. We presented each word on a picture plate containing three semantically related foils (or distractors), which reduced the chances of guessing the correct answer [57]. To reduce the likelihood of a familiarity effect with the pictures from the baseline to post testing, the pictures were randomized at post testing in their quadrant’s position on the presentation plates. We asked the children to look at the four pictures and point to the one corresponding to the target word spoken (e.g., “Show me__”). To mitigate recognition bias, we made the images used in the assessment different from those used in the program’s teaching materials. Spanish–English dual language learners completed both the English and Spanish versions, with the language administration counterbalanced. For the Spanish version, a trained member of the research team, who was fluent in Spanish, with C1 proficiency certified by the Diplomas de Español como Lengua Extranjera (DELE), administered the tool. Prior to data collection, a speech-language pathologist led a small pilot study with Head Start children (*n* = 8) to evaluate the assessment’s cultural and linguistic appropriateness.

##### Fruit and Vegetable (FV) Liking Tool (at the Baseline, Midpoints, and Post Intervention)

The researchers modified a previously validated pictorial tool for assessing children’s FV liking [33,58]. Evidence suggests the pictorial FV-liking tool is useful for nutritional educators when measuring FV liking among young children [58,59,60,61]. In a prior validation study, this measure demonstrated strong internal reliability (alpha = 0.80). Test–retest reliability (Kappa = 0.04–0.33) and concurrent validity outcomes were from weak to moderate (r_s_ = −0.25–0.26) [58]. A non-gendered five-point face scale (from super yummy to super yucky) in the tool determines the level of liking. The data collectors presented pictures, one at a time, on a university electronic device (e.g., iPad), with the scale appearing beneath the color image. We also asked the children, after they selected which face on the scale corresponded to their liking of that food, if they had eaten that food before. If the child responded “no”, the data collector selected an additional marker, “never tried”.

We adapted the tool to fit the current study. Modifications included new high-quality, digital photographs of 15 FVs to include target and non-target foods: green peas (canned), peaches (fresh and whole), bananas (fresh, whole, and partially peeled), oranges (fresh and quartered), apples (fresh and sliced), watermelon (fresh and halved), spinach (fresh and whole), tomatoes (fresh and baby), broccoli (fresh and whole), strawberries (fresh and whole), cantaloupe (fresh and halved), carrots (fresh and whole with green tops), green beans (canned and whole), green bell peppers (fresh and whole), and sweet potatoes (fresh and sliced in rounds). Prior to data collection, we cognitively evaluated the images with 103 children to ensure that the children could identify the pictured images. We selected the final images based on the children’s ability to accurately identify the foods. We chose the pictured foods because they were identified as familiar by Head Start families living in rural NC and/or had the potential to influence the SCS [33,62]. We chose the transformation of the food (sliced, whole, fresh, or canned) based on children’s ability to correctly identify the food in that state, as determined during cognitive interviews [33]. 

##### Veggie Meter^®^ (at the Baseline, Midpoints, and Post Intervention)

We measured SCSs using the Veggie Meter^®^ (Longevity Link Corp., Salt Lake City, UT, USA), a validated pressure-mediated reflection spectroscopy (RS) device. This noninvasive tool provides an objective marker of FV intake [63]. We followed the recommended protocols for data collection procedures when feasible [64,65]. After cleaning the children’s hands with soap and water or with a hand wipe, we scanned each child’s non-dominant ring finger using the single-scan mode, taking about 12 s. Because of the children’s limited attention spans and the difficulty in having them insert their finger three times into the machine, which the three-scan mode requires, we used the single-scan mode. The Veggie Meter^®^ provides a score, as derived from a spectral range of 350–850, which is assigned as the child’s SCS [66]. Previous studies have also used the single-scan method [33,64,67]. In between different children’s scans, we cleaned the lens using the optical cloth. Upon the first setup in a classroom, we allowed the Veggie Meter^®^ to acclimate to the room conditions for at least five minutes before the first assessment was taken. No recording of environmental conditions was conducted. We recalibrated the Veggie Meter^®^ after every two hours of data collection and/or after being moved to a new location. We used the same Veggie Meter^®^ for every participant at every time point. 

##### Body Mass Index (BMI) (at the Baseline, Midpoints, and Post Intervention)

We also measured the height and weight of each child, as potential co-variables for the Veggie Meter^®^ [64]. We calculated the BMI (kilograms/meter^2^) by collecting the child’s weight (to the nearest 0.10 kg) and height (to the nearest 0.10 cm), using a portable stadiometer [68] and digital bodyweight scale [69]. We calculated BMI percentiles based on the Centers for Disease Control and Prevention’s sex-specific BMI index-for-age growth charts [70].

#### 2.2.4. Data Analysis

We analyzed the data using IBM^®^ SPSS^®^ software version 29.0 [71] and SAS^®^ version 9.4 [72] and set a significance level of 0.05. We used a linear mixed model to examine child-level changes in each outcome measure from the baseline (T1) to post intervention (T4) for the Lens on Science, academic vocabulary, and the Veggie Meter^®^. The fixed effects included age, sex, and race/ethnicity, while the center was treated as a random effect. We estimated intraclass correlations from the models. Missing data patterns for the children’s scores at T1 and T4 were assessed using Little’s missing completely at random (MCAR) test [73]. Complete case analyses were performed, as no significant deviation from the MCAR assumption was detected for the outcomes. Veggie Meter^®^ scores included intermediate measurements at T2 and T3, with additional missing data; we fitted a separate linear mixed model to analyze the trends across all four time points. This model accounted for the center as a random effect and assumed an autoregressive error structure for repeated measures within the participants. We applied the Kenward–Roger degrees-of-freedom method. The fixed effects included time, age, sex, and race/ethnicity, along with interaction terms between time and demographic variables to assess subgroup differences in changes over time. We adjusted pairwise comparisons among the four time points, using Tukey’s multiple comparison procedure. We conducted descriptive statistical analyses to examine temporal trends in child FV liking ratings across the four time points (from T1 to T4). We calculated mean scores and standard deviations for each fruit, each vegetable, combined fruits, combined vegetables, and combined FV at each time point to assess patterns of the change in preference over time.

## 3. Results

A total of 273 child participants were enrolled in the study (Table 3). At the baseline, the children were male (51.6%), 3.94 (0.70) years old, Black/African American (82.1%), non-Hispanic (86.4%), and English speaking (93.8%).

### 3.1. Science Knowledge (Lens on Science)

Of the larger sample, 136 children completed the Lens on Science at the baseline (T1) and post intervention (T4). Among these, 22 (16%) were three years old, 77 (57%) four years old, and 37 (27%) five years old. The sample comprised 75 females (55%). For race/ethnicity, one hundred fifteen were Black/African American children (85%), eight (5.9%) were White, five (3.7%) were Hispanic/Latino, seven (5.2%) were multi/biracial, and one (0.7%) was another race. At the baseline, the mean scores were −0.01 (SD = 0.82), increasing to 0.33 (SD = 0.90) at T4, with an average improvement of 0.34 (SD = 0.83). Linear mixed-model analysis estimated an intraclass correlation of 0.000 and revealed a statistically significant unadjusted improvement in scores from the baseline to post intervention (mean difference = 0.34, 95% CI [0.20, 0.48], *p* < 0.001) (Figure 3). An additional mixed-model analysis identified a significant age effect (*p* = 0.024), with four-year-olds demonstrating greater improvement (mean difference = 0.41, 95% CI [0.08, 0.73]) compared to five-year-olds. However, we observed no statistically significant differences based on sex or between Black/African American children and those of other racial/ethnic backgrounds.

### 3.2. Academic Vocabulary 

Of the larger sample, 118 children completed the vocabulary assessment at the baseline (T1) and post intervention (4). Among these, 35 (30%) were three years old, 61 (52%) four years old, and 22 (19%) five years old. The sample included 60 males (51%). For race/ethnicity, eighty-eight children (75%) identified as Black/African American, eight (6.8%) were White, six (5.1%) were Hispanic/Latino, two (1.7%) were Alaska Native/American Indian, ten (8.5%) were multi/biracial, and four (3.4%) were other races. At the baseline, the mean vocabulary scores were 14.4 (SD = 4.5), increasing to 16.7 (SD = 5.3) at T4, with an average improvement of 2.3 (SD = 5.1). Linear mixed-model analysis estimated an intraclass correlation of 0.000 and revealed a statistically significant unadjusted improvement in vocabulary scores from the baseline to post intervention (mean difference = 2.3, 95% CI [1.4, 3.3], *p* < 0.001) (Figure 3). An additional mixed-model analysis identified a significant race/ethnicity effect (*p* = 0.008), indicating that Black/African American children showed, on average, 2.9 (95% CI [0.8, 5.0]), less improvement compared to children of other racial/ethnic backgrounds. However, we observed no statistically significant differences in vocabulary improvement based on sex or age group.

### 3.3. Fruit and Vegetable (FV) Liking 

Of the larger sample, 141 children completed the FV-liking tool at the baseline (T1), midpoint 1 (T2), midpoint 2 (T3), and post intervention (T4). Child-reported prior FV taste exposures were below 50% for all the FV images assessed (Figure 4 and Figure 5). Liking for all the fruits combined showed a slight decrease from the baseline (T1: M = 3.59, SD = 0.78) to post intervention (T4: M = 3.47, SD = 0.73). Similarly, liking for all the vegetables also showed a small decline from the baseline (T1: M = 2.99, SD = 0.96) to post intervention (T4: M = 2.90, SD = 1.06). When considering all the FVs combined, there was also a minor decrease in liking from the baseline (T1: M = 3.31, SD = 0.75) to post intervention (T4: M = 3.18, SD = 0.78) (Figure 6). Descriptive analyses suggested that the children’s liking for specific FVs fluctuated across the study period, with some foods becoming more preferred while others were increasingly disliked (Table 4).

### 3.4. Dietary Quality (Skin Carotenoids)

We measured 217 children with the Veggie Meter^®^ at the baseline (T1) and post intervention (T4) (Figure 7). Among these, 60 (28%) were three years old, 113 (52%) were four years old, and 44 (20%) were five years old. The sample included 113 males (52%). For race/ethnicity, one hundred seventy-six children (81%) identified as Black/African American, eleven (5.1%) were White, nine (4.2%) were Hispanic/Latino, two (0.9%) were Alaska Native/American Indian, fourteen (6.5%) were multi/biracial, and five (2.3%) were other races. At the baseline, the mean Veggie Meter^®^ score was 167 (SD = 58.1), increasing to 176 (SD = 75.7) at T4, with an average improvement of 8.7 (SD = 68.9). Linear mixed-model analysis estimated an intraclass correlation of 0.02 and found no statistically significant improvement from the baseline to post intervention (unadjusted mean difference = 6.3, 95% CI [−10.9, 23.5], *p* = 0.390) (Figure 7). An additional mixed-model analysis revealed no significant differences in Veggie Meter^®^ score changes based on race/ethnicity, sex, or age group.

We collected intermediate Veggie Meter^®^ measurements at midpoint 1 (*n* = 192) and midpoint 2 (*n* = 113), with mean scores of 189 (SD = 59.8) at T2 and 183 (SD = 66.3) at T3. A linear mixed-model analysis incorporating all four time points detected no statistically significant interactions between time and demographic variables (age, sex, and race/ethnicity). However, we observed a significant main effect of time (*p* < 0.001), indicating changes in Veggie Meter^®^ scores across the assessments. Specifically, we noted significant improvements from T1 to T2 (mean difference = 21.6, 95% CI [11.7, 31.5], *p* < 0.001) and from T1 to T3 (mean difference = 17.0, 95% CI [2.9, 31.2], *p* = 0.011). However, we observed a significant decline from T2 to T4 (mean difference = −12.8, 95% CI [−25.4, −0.3], *p* = 0.043). We adjusted these comparisons for age, sex, and race/ethnicity, using the Tukey–Kramer multiple comparison method.

## 4. Discussion

In this pilot study, we used a cross-sectional, repeated-measure research design to assess the preliminary effect of the *More PEAS Please!* program on children (3–5 years old) enrolled in Head Start. The measured child outcomes included academic and dietary outcomes by evaluating changes in science knowledge (Lens on Science), science language (Academic Vocabulary Tool), FV liking, and dietary quality (the SCS, as measured using the Veggie Meter^®^). The findings related to children’s science knowledge and language skills suggest improvement after participation in the *More PEAS Please!* program. Other studies implementing a FBL intervention in preschool children have observed positive effects on healthy food consumption [33,34,35]; however, to the best of our knowledge, this study is among the first to suggest that there may be academic benefits as well. 

Our study used the objective SCS, measured using the Veggie Meter^®^, to assess dietary quality. Although the increase from the baseline (T1) to post intervention (T4) was not statistically significant, we observed several important trends. The SCSs significantly increased from T1 (September/October 2021) to T2 (December 2021). Because the SCS reflects intake 4–6 weeks prior [63,74], this period indicates children’s intake from when they were still at home on summer break to the first few weeks of school at Head Start. The children who are in Head Start may come from families with higher rates of food insecurity compared to their more affluent counterparts [12,13,75,76], which was potentially further exacerbated by the pandemic that occurred during this study [77,78,79]. Comparatively, when in school, the children may have increased access to carotenoid-rich FVs provided through Head Start meals and snacks, which must meet the Child and Adult Care Food Program’s (CACFP’s) standards [80]. In short, the observed SCS increase from T1 to T3 may relate to the increased availability and quality of school-provided meals. Notably, the SCSs declined from T2 to T3, a period encompassing the three-week winter break. This dip aligns with prior research showing lower FV intake when children are out of school [33], reinforcing the importance of school-based access to healthy foods [81,82,83].

The most surprising trend was a significant decline from T2 to T4 (April/May 2022), during the active implementation of the 16 PEAS model science-learning activities. This could suggest that the exposure dose or intensity was insufficient to shift dietary intakes measurably or that T4 data collection occurred too soon to reflect changes, given the 4–6-week window that the SCS captures [74]. School calendars (e.g., the last day of school) constrained data collection, limiting our ability to capture longer-term changes. Future research should consider using a longitudinal study design to capture longer-term changes in children’s dietary intake, which may not be detectable within the limited time frame imposed by the school calendar.

Further complicating the interpretation, the children may still have been recovering from limited FV access during Coronavirus disease 2019 (COVID-19)-related center closures in 2020 [79,81,84,85]. This disruption may have had lingering effects on dietary habits, as meals provided at school tend to be more nutritious than those provided at home [81,82]. We also could not track individual exposure to activities or attendance because of logistical challenges, such as staffing instability and quarantine classroom closures. During the COVID-19 pandemic, Head Start centers implemented a policy mandating a one-week classroom closure if a child tested positive for COVID-19, requiring all the teachers and children to stay home for cleaning and disinfection. Any emerging positive cases during that week extended the closure to two weeks. We collected limited data from quarantined teachers. During Module 2 (January), 33.3% of the teachers reported at least one week of quarantine, with two classrooms closed for two weeks. In Module 3 (February), 13.6% of the teachers reported one week of quarantine, and one classroom closed for two weeks. The teachers reported no quarantines during Modules 4 (March) or 5 (April) [54]. Additionally, quarantine data do not reflect any additional individual classroom absences that may have occurred. As a result, we could not determine if the dosage levels were sufficient to affect the outcomes, as researchers did in prior studies [33]. Future studies should incorporate attendance tracking and use comparison groups to assess exposure-related impacts over time.

Interestingly, in this study, liking for FVs grouped individually and together, decreased across the time points. Prior research suggests that liking may decrease before increasing [33,86] and that children’s intake of healthy foods may increase before children report an increase in preference [87]. Additionally, much of the research related to repeated exposures to FVs and their impacts on child-reported liking, occurs with novel foods [40,41]. However, the children from families with lower resources may not have access to these types of foods outside of the learning environment, prompting other studies, including ours, to select more familiar target FVs for long-term sustainability for families and communities [12,13,33]. Nevertheless, promising recent findings from the 2025 Dietary Guideline Advisory Committee’s systematic review reports moderate evidence that repeated non-taste or non-taste paired with taste exposures to target FVs can increase children’s willingness to try target foods [88]. Of note, the evidence grade was higher for repeated exposures to vegetables compared to fruits [88]. 

In a previous study by Bayles and colleagues, researchers exposed children to nine familiar target vegetables through seven integrative FBL activities over four months, resulting in a similar decline in child-reported liking. Our study modified this approach by increasing the number of activities to sixteen, reducing the target vegetables to four, and extending the intervention to ten months [33]. Despite these changes, we observed a comparable decline in liking. Other studies using the same validated liking tool have observed varying results. For example, Cosco et al. conducted a randomized controlled trial with a waitlist-control design to evaluate a 13-week summer gardening intervention’s effect on children’s FV liking across 15 NC childcare centers, collecting baseline data in spring and post-intervention data in fall [59]. Although FV liking increased in the intervention group during Year 1, it declined in Year 2, while another study by Monsur et al. observed an increase in child-reported liking after a 5-week garden intervention following the same methodology as Cosco and colleagues [59,89]. Additionally, in our study, the teachers shared qualitatively that the children seemed to be tired of the repetition of the four target vegetables and recommended that additional variety, including fruits, be built into future iterations of the program [54]. Future research is needed to explore how many exposures to a familiar food are needed to impact child-reported liking and examine individual-level changes and potential moderators, such as prior exposure and dietary habits, to better understand the observed variations in FV liking [33]. Finally, the liking tool included FVs beyond the target FVs of the intervention. A recent systematic review found limited evidence that repeated taste exposures to a target vegetable may increase the acceptance of other vegetables for children aged 2–6 years [88]. However, future research is needed to better understand the effects of repeated exposures to a given set of FVs transferring to the liking of a variety of other FVs [88,90]. 

### Strengths and Limitations

The *More PEAS Please!* intervention is an innovative approach to integrate nutrition education into science and language learning to impact both school readiness and dietary quality in preschool children. This pilot study had several strengths that enhanced the credibility of the findings. The use of validated tools, such as Lens on Science and the Veggie Meter^®^, provided meaningful and objective insights into potential program impacts and strengthened the rigor of the evaluation. Additionally, conducting this pilot in the Head Start context provided valuable insights, particularly given the unique characteristics of this population and the inherent challenges of conducting assessments with young children (ages 3–5) in busy ECE settings. Working within centers that are already managing substantial programmatic and administrative responsibilities helped to test the practicality and appropriateness of the evaluation tools and PEAS components. Lastly, Head Start educators co-designed the intervention and implemented it in real-world classroom settings, which supports its feasibility and relevance in similar early childhood environments.

Although this study had notable strengths, it was not without limitations. As a pilot study, the sample size was small and drawn from a specific geographic region, which may limit generalizability to other populations. Additionally, we did not measure the outcomes in a comparison or control group to determine if the intervention or other factors caused the changes. Furthermore, tracking the individual child exposure to the intervention posed a challenge. Because of staffing shortages and pandemic-related disruptions, the children frequently transitioned between classrooms, making it difficult to assess the full dose of the program that each child received. This study was constrained by Head Start’s traditional academic calendar, which limited access to the children during the summer months. Consequently, post-intervention data (T3) were collected in May, reflecting dietary intake from March—earlier than the ideal 4–6-week post-intervention period recommended for Veggie Meter^®^ assessments [63,74].

Furthermore, because the Veggie Meter^®^ detects only colorful carotenoids, it is unable to measure colorless carotenoids or other phytochemicals found in commonly consumed healthy foods, such as apples, bananas, and berries [65,91,92]. Although the children may have increased their intake of non-carotenoid-rich FVs during the intervention, these changes would not have been captured by the device. Nonetheless, RS using the Veggie Meter^®^ remains the most objective method currently available for assessing dietary intake among young children [64]. Future studies may consider using plate-waste assessment to assess the intake of non-carotenoid-rich FVs. Future research should include more thorough tracking of exposure at the child level and incorporate comparison groups to better assess program effectiveness and dosage effects over time. Additionally, the limited time frame for the follow-up may have constrained our ability to detect longer-term dietary changes, highlighting the need for future longitudinal studies.

## 5. Conclusions

The findings from this pilot study suggest that the *More PEAS Please!* intervention may support improvements in preschoolers’ science knowledge and academic vocabulary when integrated into Head Start classrooms. Although dietary outcomes are critical, as childhood dietary behaviors have long-term effects into childhood [1,14,15,16,17,93], focusing solely on health-related outcomes may not address teachers’ perceived barriers, such as a lack of time or competing priorities (e.g., kindergarten readiness) [10,32,49,94]. The observed academic improvements in this study support the potential effectiveness of prior calls from Head Start teachers and administrators to integrate nutritional education into science learning as a strategy to enhance school readiness and foster interest in STEAM fields [27,28,32,33,34]. Although improvements in dietary outcomes were not statistically significant, trends in the SCS across the time points may highlight the potential influence of an intervention coupled with school-based food access and exposure, particularly in programs that participate in CACFP [80]. Lastly, the lessons learned from this pilot study will be used to further refine the *More PEAS Please!* program materials, implementation, and evaluation strategies. Collectively, our findings highlight the promise of integrative FBL for promoting academic outcomes and healthy eating behaviors. As ECE programs face growing pressure to positively impact both academic and health outcomes, FBL programs, like *More PEAS Please!*, may offer a promising, integrative approach to effectively accomplish both goals.

## Figures and Tables

**Figure 1 nutrients-17-01523-f001:**
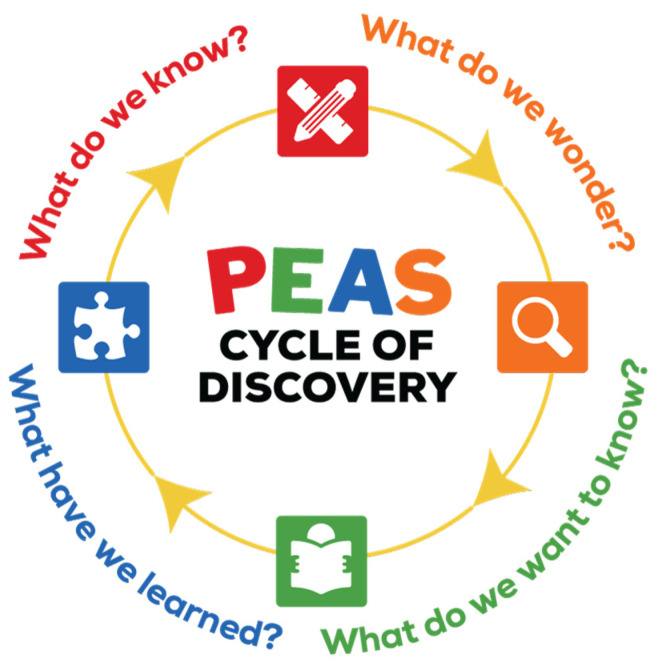
*More PEAS Please!* cycle of discovery.

**Figure 2 nutrients-17-01523-f002:**
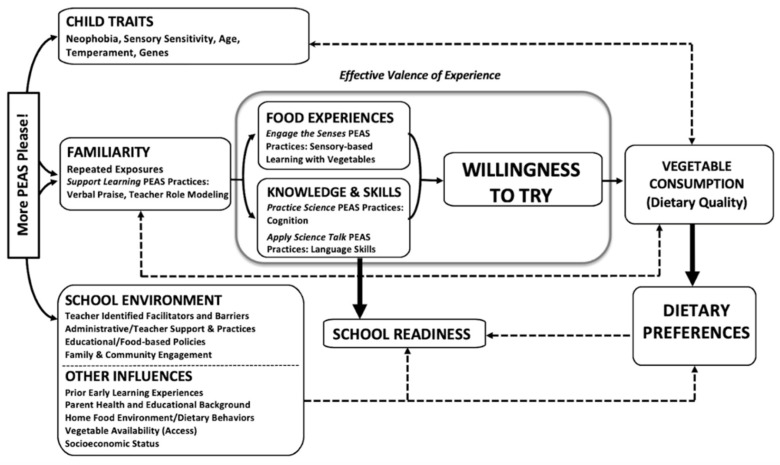
*More PEAS Please!* child-level theoretical framework.

**Figure 3 nutrients-17-01523-f003:**
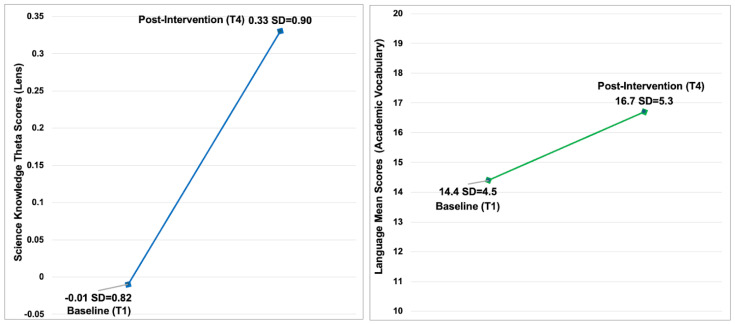
Science knowledge (Lens on Science) (*n* = 136) (**left**) and language development (academic vocabulary) (*n* = 118) (**right**) at the baseline and post intervention. The linear mixed models for science knowledge (unadjusted mean difference = 0.41; 95% CI [0.17, 0.50], *p* < 0.001, intraclass correlation = 0.000) and language development (unadjusted mean difference = 2.3; 95% CI [1.4, 3.3], *p* < 0.001, intraclass correlation = 0.000) significantly improved between the baseline (T1) and post intervention (T4).

**Figure 4 nutrients-17-01523-f004:**
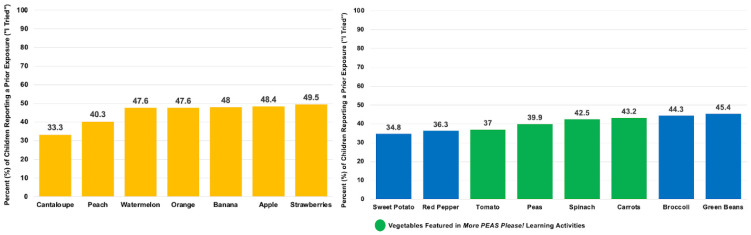
Child-reported exposure to fruits and vegetables (FVs) (*n* = 141) at the baseline. Fruits are depicted by the yellow bars; vegetables are depicted by the blue and green bars. The children were asked to self-report their prior taste exposure during the baseline assessment of the FV liking. The FVs targeted in the intervention (green bars) included carrots, tomatoes, spinach, and peas.

**Figure 5 nutrients-17-01523-f005:**
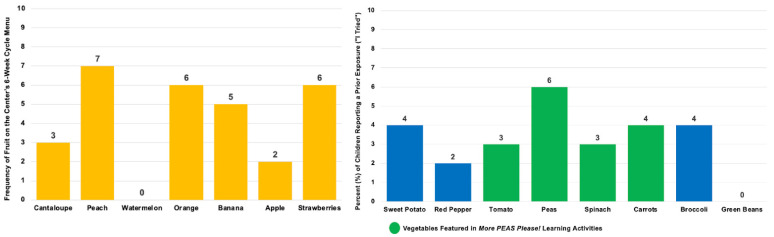
Frequency of FVs occurring on the center’s menus over a six-week period. Fruits are depicted by the yellow bars; vegetables are depicted by the blue and green bars. Notes: Juices (e.g., orange and apple) were not included in the counts. The FVs targeted in the intervention (green bars) included carrots, tomatoes, spinach, and peas.

**Figure 6 nutrients-17-01523-f006:**
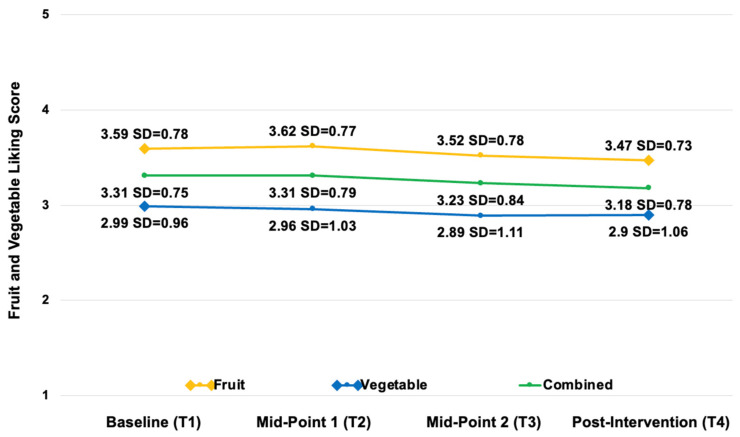
Child-reported FV likings at the baseline, midpoint 1, midpoint 2, and post intervention (*n* = 141). Scale: 1, super yucky; 5, super yummy. In between T2 and T3, the children were absent from school for 3 weeks for winter break; however, the duration between the data collection points was approximately equal.

**Figure 7 nutrients-17-01523-f007:**
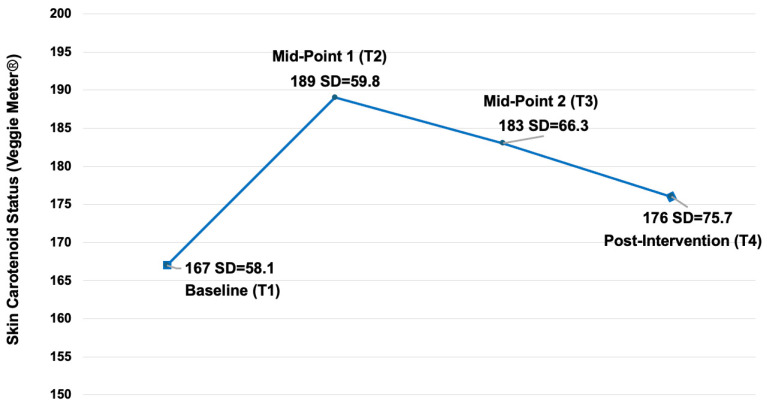
The skin carotenoid statuses (SCSs) at the baseline, midpoint 1, midpoint 2, and post intervention (*n* = 217). Scale = 0−850 for Veggie Meter^®^ (Longevity Link Corporation, Salt Lake City, UT, USA) scores. In between T2 and T3, the children were absent from school for 3 weeks for winter break; however, the duration between the data collection points was approximately equal. Linear mixed-model results: from T1 (Baseline) to T2 (Midpoint 1) the mean difference = 21.6, 95% CI [11.7, 31.5], *p* < 0.001; from T1 (Baseline) to T3 (Midpoint 2), the mean difference = 17.0, 95% CI [2.9, 31.2], *p* = 0.011; from T1 (Baseline) to T4 (Post Intervention), the unadjusted mean difference = 6.3, 95% CI [−10.9, 23.5], *p* = 0.390, intraclass correlation = 0.02.

**Table 1 nutrients-17-01523-t001:** *More PEAS Please!* classroom model science-learning activities.

Unit Topic *	Learning Objectives and Activities	Food Exposure
Living things ^ad^	Children will be able to identify living and nonliving things and make comparisons between the two by observing observable characteristics.	Carrots, peas, tomatoes, and spinach
Seeds ^ad^	Children will be able to identify seeds as living things and describe their needs (light, water, space, food, temperature, and soil).	Peas and tomatoes
Plants ^abd^	Children will be able to identify plants as living things and describe their needs (light, water, space, food, and soil).	Spinach, carrots, peas, tomatoes, and spinach
Plant parts ^cd^	Children will be able to describe the structures and functions of plant parts (i.e., roots, stems, and leaves).	Carrots and spinach

* The model science-learning activities were organized into four thematic life science units. Following the *Preschool Cycle of Discovery*, each unit consisted of four hands-on activities (What do we know? What do we wonder? What do we want to know? What have we learned?). Next-generation science standards (NGSS), disciplinary core idea (life science) alignment: ^a^ LS1.C organization for matter and energy flow in organisms: How do organisms obtain and use the matter and energy they need to live and grow? ^b^ LS2.A interdependent relationships in ecosystems: How do organisms interact with living and nonliving environments to obtain matter and energy? ^c^ LS1.A structure and function: How do the structures of organisms enable life’s functions? ^d^ Interdependence of science, engineering, and technology.

**Table 2 nutrients-17-01523-t002:** *More PEAS Please!* cycle of discovery and science practice alignment.

Cycle of Discovery Focus Within Each Unit	Next-Generation Science Standards (NGSS) Practice(s)
**What Do We Know? (Activity 1):** Teachers begin a unit by exploring what children know about the topic in a large group discussion. Children are prompted to describe what they know about a science concept (e.g., What do seeds look like?), using their five senses to explore the characteristics of the observed thing (e.g., some seeds are big and some are round). Teachers represent children’s knowledge through conversation, interaction, and creating a visual documentation of children’s responses.	**Practice 1:** Asking questions and defining problems; **Practice 2:** Developing and using models
**What Do We Wonder? (Activity 2):** Teachers engage children in an experiment to explore an observable science phenomenon (e.g., seeds germinating) related to the unit topic. Children explore the science concept by learning how to ask questions that can be investigated (e.g., what does a seed need to germinate or “start to grow”?), make predictions, conduct experiments, make observations, and report and discuss their findings. Children also formulate and check predictions through observations and experimentation (with adult support and modeling) and use language and vocabulary to describe the object’s characteristics and attributes.	**Practice 2:** Developing and using models; **Practice 3:** Planning and carrying out investigations; **Practice 4:** Analyzing and interpreting data
**What Do We Want to Know? (Activity 3):** Teachers and children share what they have learned so far, ask new questions, and read a book related to the topic (e.g., “*Seeds, Seeds, Seeds*” by Nancy Elizabeth Wallace). With teacher or family support, children explain what they have learned and ask new questions. They ask more questions and identify ways to find answers (e.g., look in a book, use the computer, and observe).	**Practice 2:** Developing and using models; **Practice 6:** Constructing explanations and designing solutions
**What Have We Learned? (Activity 4):** Finally, on the last day of a unit, teachers review with the children what the class learned and provide a tasting experience. Children integrate and reflect on what they learned about the science concept (e.g., “There are many kinds of seeds. Seeds need water to germinate. The wind, animals, and bodies of water can help seeds to move”).	**Practice 2:** Developing and using models; **Practice 6:** Constructing explanations and designing solutions; **Practice 7:** Engaging in argument from evidence; **Practice 8:** Obtaining, evaluating, and communicating information

NGSS indicates next-generation science standards.

**Table 3 nutrients-17-01523-t003:** Child demographics at the baseline and retention rates for data collection measures at each time point (*n* = 273).

Characteristic	*n* (%)	Mean	SD
**Age**	—	3.94	0.70
**Sex**			
Male	141(51.6)	–	–
Female	132 (48.4)	–	–
**Race** **^a^**			
Black/African American	224 (82.1)	—	—
Hispanic	12 (4.4)	—	—
White	12 (4.4)	—	—
Asian	2 (0.7)	—	—
Other	22 (8.0)	—	—
**Ethnicity** **^b^**			
Non-Hispanic/Latino	236 (86.4)	—	—
Hispanic/Latino	35 (12.8)	—	—


	**Baseline (*n*)**	**Post Intervention (*n*)**	**Retention ^c^ (%)**
LENS on Science	202	136	67.33
Academic Vocabulary	200	118	59.0
FV Liking	141	103	73.05
SCS	273	217	79.49

SD indicates standard deviation; SCS indicates the skin carotenoid status. ^a^ Missing data for race, *n* = 1. ^b^ Missing data for ethnicity, *n* = 2. ^c^ The retention rates in the table reflect the children’s participation in data collection measures at the baseline and post intervention. The values reported for each measurement are independent.

**Table 4 nutrients-17-01523-t004:** Child-reported likings by individual FVs at the baseline, midpoint 1, midpoint 2, and post intervention (*n* = 141).

	Baseline 1	Midpoint 1	Midpoint 2	Post Intervention
	Mean (SD)	Mean (SD)	Mean (SD)	Mean (SD)
**Fruit**				
Cantaloupe	2.57 (1.74)	2.30 (1.59)	2.15 (1.63)	2.24 (1.67)
Peaches	3.17 (1.66)	3.33 (1.74)	2.91 (1.81)	3.28 (1.72)
Watermelon	4.04 (1.25)	4.16 (1.24)	4.19 (1.31)	3.86 (1.49)
Oranges	3.92 (1.31)	4.05 (1.35)	4.25 (1.29)	4.16 (1.18)
Bananas	4.01 (1.27)	4.05 (1.33)	4.0 (1.52)	4.01 (1.35)
Apples	3.77 (1.38)	4.09 (1.27)	3.78 (1.58)	3.69 (1.55)
Strawberries	4.21 (1.14)	4.20 (1.21)	4.47 (0.94)	4.31 (1.06)
**Vegetable**				
Sweet Potatoes ^a^	2.84 (1.68)	2.93 (1.68)	2.84 (1.71)	2.71 (1.74)
Red Peppers	2.58 (1.63)	2.46 (1.63)	2.48 (1.70)	2.30 (1.60)
Tomatoes ^a^	2.95 (1.68)	2.69 (1.74)	2.46 (1.66)	2.36 (1.67)
Peas ^a^	2.93 (1.70)	2.78 (1.75)	2.76 (1.77)	2.55 (1.74)
Spinach ^a^	2.93 (1.67)	2.87 (1.70)	2.79 (1.82)	2.91 (1.79)
Carrots ^a^	3.16 (1.65)	3.20 (1.71)	3.12 (1.80)	3.12 (1.74)
Broccoli	3.22 (1.63)	3.30 (1.72)	2.85 (1.82)	3.17 (1.76)
Green Beans	3.44 (1.59)	3.15 (1.63)	3.40 (1.77)	3.42 (1.66)

SD indicates standard deviation. ^a^ The FVs targeted in the intervention included carrots, tomatoes, spinach, and peas.

## Data Availability

The data presented in this study are available upon request from the corresponding author. Because of privacy concerns, they are not publicly available.

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
