# Peer review of "A Food-Based Science, Technology, Engineering, Arts, and Mathematics Learning Program May Improve Preschool Children’s Science Knowledge and Language Skills in Rural North Carolina"

_nutrients, 2025, doi:10.3390/nu17091523_

Round 1

Reviewer 1 Report

Comments and Suggestions for Authors

Summary
This is an interesting pilot study that uses a food-based intervention program to encourage healthy eating while improving academic performance. The study was carefully planned and implemented. As an educator of both nutrition and science, I found this to be an inspiring study.  The manuscript is well written and provides new information regarding potential interventions for improved health and academic achievement in young children. I offer a few suggestions for improvements below.

Title
Since the title is long, I suggest removing “Pilot Study” and “(3-5 years)” since these details are mentioned in the abstract. Or perhaps reword “Children (3-5 years) to “Preschool Children”.

Abstract
In line 28 and 362, I assume “(0.70)” refers to standard deviation, but this is not clear. I suggest changing this to 3.94 +/- 0.70 years old or 3.94 (SD=0.70) years old.

Introduction
 Provides a well-written comprehensive background of the topic. I suggest moving the study aims and hypothesis to this section (see comment below).

Materials and Methods
I found it unusual to see the study aims listed under the materials and methods section and would suggest moving it to the end of the introduction section with a hypothesis included. Although a hypothesis can be found in line 214 within the methods section, I believe most readers will be looking for this at the end of the introduction. In Table 2, the second column is difficult to read due to the formatting, but this should be an easy fix for the author or editor. 

Results
In line 27 and 382 you note that participants are mostly male, although the split between male and female seems fairly balanced at 51.6% male and 48.4% female. In Tables 3 and 4, it seems unnecessary to abbreviate the term “mean” when it’s a small word and you have plenty of space in the cell. I would just write the whole word. In table 4, the abbreviations need to be added as a footnote. In section 3.1 Science Knowledge (Lens on Science), you start by providing percentages for each age, but then only provide percentages for one gender and one race. I suggest including percentages for each subgroup. If you only desire to list the subgroup that is in the highest percentage, as you did for Black/African American children (85%), then you should list females (55%) rather than males (45%) in this section. I understand that you likely only mention males here to be consistent with your results that follow, but highlighting only males repeatedly creates a perception that the females are not a priority in this study.

Discussion
It may be worth mentioning that since the Veggie Meter only measures carotenoids, other common phytochemicals, such as polyphenols, cannot be measured. Apples, bananas, and strawberries are common fruits for young children, but are lower in carotenoids and higher in polyphenols (and other classes of phytochemicals). It is possible that these preschool participants may have increased their intake of these fruits during the study, but this would have gone unnoticed due to the limitations of the instrument. 

Author Response

Comment: This is an interesting pilot study that uses a food-based intervention program to encourage healthy eating while improving academic performance. The study was carefully planned and implemented. As an educator of both nutrition and science, I found this to be an inspiring study.  The manuscript is well written and provides new information regarding potential interventions for improved health and academic achievement in young children. I offer a few suggestions for improvements below.

Response: Thank you for taking the time to review our manuscript and provide helpful feedback. We believe your review has enhanced the quality of our manuscript.

Comment: Since the title is long, I suggest removing “Pilot Study” and “(3-5 years)” since these details are mentioned in the abstract. Or perhaps reword “Children (3-5 years) to “Preschool Children”.

Response: Thank you for this comment. We agree that the title is long. We have removed the words' Pilot Study” since they appear early in the abstract. We also revised “Children (3-5 years) to “Preschool Children” as suggested. See Lines 2-3 for revision.

Comment: In line 28 and 362, I assume “(0.70)” refers to standard deviation, but this is not clear. I suggest changing this to 3.94 +/- 0.70 years old or 3.94 (SD=0.70) years old.

Response: Thank you for this comment; this revision has been completed. See Line 31 for revision.

Comment: Provides a well-written comprehensive background of the topic. I suggest moving the study aims and hypothesis to this section (see comment below).

Response: We agree, this revision has been made. See Lines 125-131 for revision.

Comment: I found it unusual to see the study aims listed under the materials and methods section and would suggest moving it to the end of the introduction section with a hypothesis included. Although a hypothesis can be found in line 214 within the methods section, I believe most readers will be looking for this at the end of the introduction. 

In Table 2, the second column is difficult to read due to the formatting, but this should be an easy fix for the author or editor. 

Response: Thank you for this feedback. We agree that adding our hypothesis for the study is essential. We have made this revision. See Lines 128-130 for revision. Table 2 has also been revised to improve readability. We left-aligned the next and adjusted the spacing to reduce the number of hyphenated words. We hope these changes improve overall readability. See Table 2, Line 185 for revisions.

Comment: In line 27 and 382 you note that participants are mostly male, although the split between male and female seems fairly balanced at 51.6% male and 48.4% female. 

In Tables 3 and 4, it seems unnecessary to abbreviate the term “mean” when it’s a small word and you have plenty of space in the cell. I would just write the whole word. In table 4, the abbreviations need to be added as a footnote. 

In section 3.1 Science Knowledge (Lens on Science), you start by providing percentages for each age, but then only provide percentages for one gender and one race. I suggest including percentages for each subgroup. If you only desire to list the subgroup that is in the highest percentage, as you did for Black/African American children (85%), then you should list females (55%) rather than males (45%) in this section. I understand that you likely only mention males here to be consistent with your results that follow, but highlighting only males repeatedly creates a perception that the females are not a priority in this study.

Response: Thank you for your comments. We have addressed each in order below.

We have removed the word “mostly” from the results text describing gender. See Line 366 for the revision.

In Tables 3 and 4, we have spelled out the word Mean. In Table 4, we have added “SD indicates standard deviation” as a footnote. Refer to Tables 3 and 4 for the revisions.

These are excellent points. 

We have reported the number/percent of females in text versus males; since this variable is dichotomous, we did not report both. Additionally, the race/ethnicity categories have been expanded for each sample of the outcomes. Refer to Lines 380-382, 403-405, and 449-451 for the revisions.

Comment: It may be worth mentioning that since the Veggie Meter only measures carotenoids, other common phytochemicals, such as polyphenols, cannot be measured. Apples, bananas, and strawberries are common fruits for young children, but are lower in carotenoids and higher in polyphenols (and other classes of phytochemicals). It is possible that these preschool participants may have increased their intake of these fruits during the study, but this would have gone unnoticed due to the limitations of the instrument.

Response: Thank you for this comment. We agree and have added new text to reflect this limitation.  Refer to Lines 594-606 for the revisions.

Reviewer 2 Report

Comments and Suggestions for Authors

I believe this manuscript is relevant to the addressed topics and it can be considered for publication in Nutrients after some revisions. These are my suggestions:

In the abstract, before the study’s aims, you need to provide a background statement.

Your study’s novelty needs to be better explained in the Introduction. At the end of this section, the study’s objectives need to be clarified. These study’s objectives shouldn’t be placed in the Materials and Methods section.

The description of the More PEAS Please! Program is too brief. More information is required so the readers can understand well the goals of this program.

Where is the information about the ethics procedures? You have to provide it inside your manuscript.

The Results are adequately described, but the Discussion can be improved. Why are your obtained results relevant to the worldwide scientific community and populations? More similar studies should be analyzed.

You shouldn’t have citations in your Conclusions. This section can also be improved, and further directions for future investigations should be given.

Author Response

Comment: I believe this manuscript is relevant to the addressed topics and it can be considered for publication in Nutrients after some revisions. These are my suggestions: In the abstract, before the study’s aims, you need to provide a background statement.

Response: Thank you for taking the time to review our manuscript and provide helpful feedback. We have added a background statement to the beginning of the abstract. 

Comment: Your study’s novelty needs to be better explained in the Introduction. At the end of this section, the study’s objectives need to be clarified. These study’s objectives shouldn’t be placed in the Materials and Methods section.

Response: Thank you for this feedback. We agree that adding our hypothesis for the study is essential. We have made this revision by moving the study objective. See Lines 125-131 for revision. Refer to lines 79-119 for a description of how the study aligns with current understanding. The novelty of our study is described on Lines 120-124.

Comment: The description of the More PEAS Please! Program is too brief. More information is required so the readers can understand well the goals of this program.

Response: The program description is currently three pages in length. The paragraph on Lines 134-142 provides an overview of the program. The specific program goal can be found on Lines 136-138. Additional information is provided through detailed descriptions of the program components at both the teacher and child levels (Sections 2.1.1 and 2.2.2). Tables and Figures are provided to provide additional information regarding specific curricular content. Finally, Section 2.1.4 provides a detailed explanation and theoretical model for the driving child-level behavioral and program theory. Considering the length and detail of the current program description, the authors would need more specific guidance on what additional information is needed to help readers understand the program goals before making further revisions.

Comment: Where is the information about the ethics procedures? You have to provide it inside your manuscript.

Response: Per journal guidance, the IRB statement is provided on Lines 647-649.

Comment: The Results are adequately described, but the Discussion can be improved. Why are your obtained results relevant to the worldwide scientific community and populations? More similar studies should be analyzed.

Response: As previously described in the introduction, more similar studies are not described because the research in this area is limited. We reviewed the primary studies available in the introduction. All possible comparisons were made in the discussion. The conclusion statement is relevant to international communities. Our team believes it would be premature to extend this statement further, considering the limited research and the fact that this study was a pilot and did not include a comparison group.

Comment: You shouldn’t have citations in your Conclusions. This section can also be improved, and further directions for future investigations should be given.

Response: To our knowledge, the journal guidelines do not restrict the inclusion of citations in the conclusions. See Lines 626-627 for the next steps. We also include future research within our discussion, as contextually it made more sense for these statements to be made closer to the relevant discussion points. We also modeled after other similar articles already published in Nutrients. If the Editor and Reviewer feel strongly that these statements should be moved to the conclusion, we are happy to make this revision.